# Body mass index and adipose distribution have opposing genetic impacts on human blood traits

Christopher S Thom[1]*, Madison B Wilken[1], Stella T Chou[2], Benjamin F Voight[3,4,5]*

[1]Division of Neonatology, Children's Hospital of Philadelphia, Philadelphia, United States; [2]Division of Hematology, Children's Hospital of Philadelphia, Philadelphia, United States; [3]Department of Systems Pharmacology and Translational Therapeutics, University of Pennsylvania - Perelman School of Medicine, Philadelphia, United States; [4]Department of Genetics, University of Pennsylvania - Perelman School of Medicine, Philadelphia, United States; [5]Institute for Translational Medicine, University of Pennsylvania - Perelman School of Medicine, Philadelphia, United States

**Abstract** Body mass index (BMI), hyperlipidemia, and truncal adipose distribution concordantly elevate cardiovascular disease risks, but have unknown genetic effects on blood trait variation. Using Mendelian randomization, we define unexpectedly opposing roles for increased BMI and truncal adipose distribution on blood traits. Elevated genetically determined BMI and lipid levels decreased hemoglobin and hematocrit levels, consistent with clinical observations associating obesity and anemia. We found that lipid-related effects were confined to erythroid traits. In contrast, BMI affected multiple blood lineages, indicating broad effects on hematopoiesis. Increased truncal adipose distribution opposed BMI effects, increasing hemoglobin and blood cell counts across lineages. Conditional analyses indicated genes, pathways, and cell types responsible for these effects, including *Leptin Receptor* and other blood cell-extrinsic factors in adipocytes and endothelium that regulate hematopoietic stem and progenitor cell biology. Our findings identify novel roles for obesity on hematopoiesis, including a previously underappreciated role for genetically determined adipose distribution in determining blood cell formation and function.

*For correspondence:
thomc@chop.edu (CST);
bvoight@pennmedicine.upenn.edu (BFV)

Competing interest: The authors declare that no competing interests exist.

## Editor's evaluation

The study shows that genetically determined adiposity plays a previously underappreciated role in determining blood cell formation and function. The authors have clearly spelled out their hypothesis and performed all the relevant and available analyses in the "Mendelian Randomization toolbox". The study will help understand the pathogenesis for clonal hematopoiesis.

## Introduction

Blood cell homeostasis is achieved through incompletely understood coordination of blood cell-intrinsic gene regulation and blood cell-extrinsic environmental mechanisms (*Comazzetto et al., 2021*; *Ulirsch et al., 2019*). The importance of blood cell formation and function in normal hematopoietic development, hematologic diseases, and clinical manifestations of systemic disorders has prompted extensive investigation of loci underlying human blood trait variation through genome-wide association studies (GWAS) (*Astle et al., 2016*; *Chen et al., 2020*; *Vuckovic et al., 2020*). One shortcoming has been an inability to identify extrinsic effects from these data.

Adipocytes and endothelial cells within the bone marrow environment regulate hematopoiesis (*Comazzetto et al., 2021*; *Zhong et al., 2020*). Discrete adipocyte populations differentially modulate systemic physiology and homeostasis (*Hildreth et al., 2021*). For example, white adipose tissue has a derogatory effect on hematopoiesis, whereas mesenchymal-derived bone marrow adipocyte populations support blood cell formation (*Comazzetto et al., 2021*; *Cuminetti and Arranz, 2019*; *Wang et al., 2018*; *Zhong et al., 2020*).

Single nucleotide polymorphisms (SNPs) that genetically increase body mass index (BMI) also raise metabolic and cardiovascular disease risks (*Pulit et al., 2019*). Some observational studies have linked obesity (BMI > 30 kg/m$^2$; *Aigner et al., 2014*) or hypercholesterolemia (*Shalev et al., 2007*) with anemia. However, others observed apparent erythrocytosis in obese individuals (*Keohane et al., 2013*). Genetic relationships have not been elucidated for BMI on erythroid or other blood traits. A genetic predisposition to accumulate truncal adipose tissue elevates waist-to-hip ratio (WHR). Like BMI, WHR influences cardiovascular risks (*Huang et al., 2021*; *Pulit et al., 2019*), but genetic impacts of WHR on blood traits are unknown.

Our study was designed to test several hypotheses. First, we wanted to determine if increased BMI decreased hemoglobin (HGB) at the genetic level, consistent with higher anemia risk in obese individuals (*Aigner et al., 2014*). Second, we wanted to identify BMI-related traits and mechanisms responsible for effects on erythroid traits (e.g., HGB level and hematocrit [HCT]). We specifically examined effects of adipose distribution (WHR), which impacts cardiovascular disease risk along with BMI variation (*Huang et al., 2021*). We also analyzed genetically determined lipid fractions, since hyperlipidemia has been linked with anemia risk (*Shalev et al., 2007*) and since lipids can impact erythrocyte stability (*Mohandas and Gallagher, 2008*). Third, we asked if BMI and related traits impacted nonerythroid blood cell lineages. This led us to assess genetic impacts on a total of 15 quantitative blood traits, which generally reflect perturbations in blood cell formation and/or function (*Chen et al., 2020*; *Vuckovic et al., 2020*).

We used a Mendelian randomization (MR) framework for our study, anticipating that the results would help to clarify the complex interplay between cardiometabolic traits and hematopoiesis without necessarily revealing clinically apparent effects. MR leverages variants linked to an exposure trait to estimate causal genetic effects on an outcome (*Hemani et al., 2018*). Multivariable MR (MVMR) and causal mediation analyses can parse effects from multiple factors (*Burgess et al., 2017*). Interrogating causal effects of BMI and related factors on erythroid and other blood traits revealed unexpected associations between BMI, WHR, and hematopoietic variation. Conditional genome-wide analyses using mtCOJO (*Zhu et al., 2018*) highlighted blood loci that were substantially influenced by BMI and/or WHR, helping to reveal genes and pathways by which these physiological factors impact blood trait variation.

## Results

### Causal association between genetically determined BMI and lower HGB level

We hypothesized that BMI would decrease HGB at the genetic level, consistent with clinical observations. Using MR, we found that each standard deviation (SD) unit increase in BMI caused a 0.057 SD decrease in HGB levels by the inverse variance weighted (IVW) method (p = 1.0 × 10$^{-5}$) that was directionally consistent across sensitivity analyses without evidence of horizontal pleiotropy or weak instrument bias (*Figure 1a*, *Figure 1—figure supplement 1a*, *Supplementary file 1*—Table 1). Similar effects were observed for HCT (*Figure 1b* and *Figure 1—figure supplement 1b*), suggesting BMI is genetically linked with reduced HGB.

### Cholesterol levels impact erythroid traits independent of BMI

We next investigated previously proposed mechanisms to explain observational links between BMI and anemia. For example, we hypothesized that hypercholesterolemia may cause anemia (*Shalev et al., 2007*) through altered erythrocyte membrane formation and stability (*Mohandas and Gallagher, 2008*). We confirmed that increased total cholesterol (TC) or lipid fractions (low density lipoprotein [LDL] or high density lipoprotein [HDL]), but not triglyceride levels (TG), decreased HGB or HCT (*Figure 1—figure supplements 2–3*). However, multivariable and mediation experiments revealed

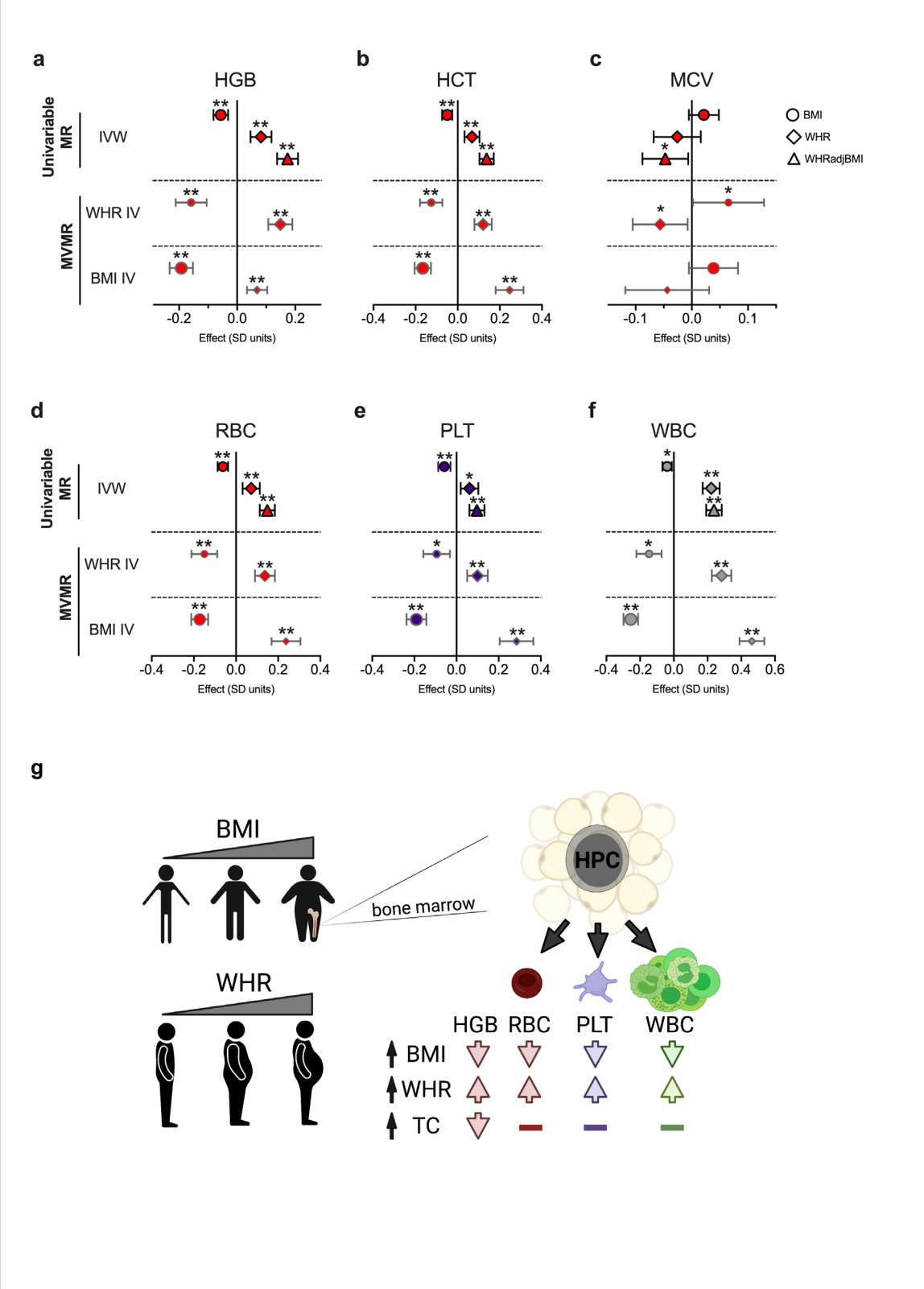

**Figure 1.** Body mass index (BMI) and waist-to-hip ratio (WHR) exert opposing effects on blood traits. (a–f) Effects of BMI, WHR, WHRadjBMI on (a) hemoglobin (HGB), (b) hematocrit (HCT), (c) mean corpuscular volume (MCV), (d) red blood cell count (RBC), (e) platelet count (PLT), or (f) white blood cell count (WBC). Shown in top panel are effects of BMI, WHR, or WHRadjBMI on HGB in univariable Mendelian randomization (MR) experiments by inverse variance weighted (IVW) method. Underneath univariable MR results, effects of BMI or WHR at 639 LD-independent WHR-associated single

*Figure 1 continued on next page*

*Figure 1 continued*

nucleotide polymorphisms (SNPs) are shown. Bottom row of panels show effects of BMI or WHR at 1268 LD-independent BMI-associated SNPs. Effects are in SD units with 95% confidence intervals. *p < 0.05, **p < 0.003. (**g**) Schematic summarizing effects of indicated exposures on blood traits (created with https://BioRender.com).

The online version of this article includes the following figure supplement(s) for figure 1:

**Figure supplement 1.** Genetically determined body mass index (BMI) decreases hemoglobin (HGB) and hematocrit (HCT) levels.

**Figure supplement 2.** Effects of lipid fractions or triglyceride level (TG) on erythroid traits.

**Figure supplement 3.** Total cholesterol (TC) decreases hemoglobin (HGB) and hematocrit (HCT) levels independent of body mass index (BMI) effects.

**Figure supplement 4.** Effects of body mass index (BMI), waist-to-hip ratio (WHR), and WHRadjBMI on mean corpuscular volume (MCV) across Mendelian randomization (MR) methodologies.

**Figure supplement 5.** Effects of hemoglobin (HGB) or hematocrit (HCT) on body mass index (BMI).

**Figure supplement 6.** Effects of waist-to-hip ratio (WHR) and WHRadjBMI are consistent across Mendelian randomization (MR) methodologies.

**Figure supplement 7.** Genetically determined waist-to-hip ratio (WHR) and body mass index (BMI) exert opposing effects on multilineage quantitative blood traits, including red blood cell (RBC), platelet (PLT), and white blood cell (WBC) count.

**Figure supplement 8.** Effects of body mass index (BMI) and waist-to-hip ratio (WHR) on quantitative blood traits by multivariable Mendelian randomization (MVMR).

**Figure supplement 9.** Effects of waist-to-hip ratio (WHR) and body mass index (BMI) on quantitative blood traits by multivariable Mendelian randomization (MVMR) after regressing out effects of other blood traits.

that cholesterol levels alter erythroid traits via mechanisms independent from BMI (*Figure 1—figure supplement 3b* and *Supplementary file 1*—Table 2). Chronic inflammation and iron deficiency, which cause decreased erythrocyte size (microcytosis), have also been suggested to mediate obesity-related anemia (*Aigner et al., 2014*). However, BMI did not alter erythrocyte mean corpuscular volume (MCV) by MR (p = 0.12, *Figure 1c*, *Figure 1—figure supplement 4a*). In sum, these findings aligned with clinical observations linking BMI with anemia risk, but argued against prevailing mechanistic hypotheses at the genetic level.

Reverse causality experiments also identified inverse correlations between erythroid and metabolic traits (*Figure 1—figure supplement 5*). Directional MR Steiger (*Hemani et al., 2017*) analyses were inconsistent (*Supplementary file 1*—Table 3), perhaps limited by blood trait measurement variation or quantitative adjustments for individual characteristics (*Vuckovic et al., 2020*).

## Genetic impacts of WHR oppose BMI effects on blood trait variation

We then considered an alternative hypothesis that the physiological distribution of adipose, as measured by WHR, could impact BMI-related anemia risk. Unexpectedly, and in contrast to BMI, higher WHR increased red blood cell (RBC) traits (*Figure 1a and b* and *Figure 1—figure supplement 6a*,b). WHR adjusted for BMI on an individual level (WHRadjBMI) exacerbated these positive effects (*Figure 1a and b* and *Figure 1—figure supplement 6c*,d) and also associated with decreased MCV (*Figure 1c* and *Figure 1—figure supplement 4b*,c). Multivariable analyses formally validated the opposing, cross-mediating effects of BMI and WHR on erythroid traits (*Figure 1a, b and c*).

Next, we asked whether BMI- and WHR-related effects impacted non-erythroid blood traits. Multivariable and mediation analyses on other blood traits identified cross-mediating opposing effects of WHR and BMI on quantitative blood counts across cell lineages (*Figure 1d, e and f* and *Figure 1—figure supplements 7–8*). These effects persisted after accounting for related blood traits (*Figure 1—figure supplement 9*). The directionally consistent effects across multiple lineages suggested that underlying mechanisms related to hematopoietic stem and progenitor cells (HSCs) common to these lineages (*Thom and Voight, 2020*; *Figure 1g*). These findings also argued against sex-related effects. Men generally have higher HGB (*Vuckovic et al., 2020*) and WHR (*Pulit et al., 2019*), but women can have higher platelet and neutrophil counts (*Bain, 1996*).

## BMI- and WHR-adjusted conditional analyses clarify blood trait variation loci

Our findings suggested that effects from BMI and WHR were likely to have influenced some previously reported blood trait loci. To identify blood loci related to these factors, we applied mtCOJO

(*Zhu et al., 2018*) to condition blood trait GWAS data on polygenic-measured BMI and/or WHR. BMI adjustment modestly changed SNP effect sizes and retained most (91%) unadjusted lead sentinel variants (*Figure 2a and b* and *Supplementary file 1*—Table 4). However, combined BMI and WHR adjustment resulted in substantial SNP effect size changes (>36-fold increased SD (SD = $7.6 \times 10^{-3}$) vs. BMI adjustment alone (SD = $2.1 \times 10^{-4}$), p < 0.0001 by F-test to compare variances), with a negative skew (–2.0) reflecting adjustment for the positive effect of WHR on HGB levels (*Figure 2a and b*). Thus, while prior GWAS adjusted for BMI (*Astle et al., 2016*), we identified more prominent effects for WHR across blood traits and lineages.

Combined BMI/WHR adjustment shifted 341 HGB-associated loci toward the null, supporting a key role for BMI- and WHR-mediated mechanisms at these sites, while also identifying 844 sites that either clarified interpretation of previously implicated HGB loci or tagged previously unreported regions (n = 242 novel loci, *Figure 2b* and *Supplementary file 1*—Table 5). For example, a missense coding variant in *RAPGEF3* (rs145878042), previously linked to BMI (*Pulit et al., 2019*), WHRadjBMI (*Pulit et al., 2019*), and platelet distribution width (*Vuckovic et al., 2020*), did not meet genome-wide significance for HGB (p = 0.003) until BMI/WHR adjustment (p = $2.4 \times 10^{-14}$, *Figure 2b and c*). Interpretation of SNPs at the *RSPO3* locus also dramatically changed (*Figure 2b and d*). *RSPO3*, a Wnt pathway modulator that directs development of bone and other tissues (*Nilsson et al., 2021*), has been linked with adipose distribution (*Pulit et al., 2019*) and blood trait variation (*Vuckovic et al., 2020*). Similar effects were seen in adjusted HCT data (*Figure 2a and b* and *Supplementary file 1*—Tables 6–7) and quantitative traits across blood lineages (*Figure 2—figure supplement 1*). These conditional analyses presumably revealed sites where BMI and/or WHR biology most strongly impact blood trait variation, although it is possible that some pleiotropic loci independently regulate blood traits through shared or different gene regulation.

Functional enrichment analyses identified many consistent genes and processes in unadjusted vs. BMI/WHR-adjusted data across blood traits, with some notable changes (*Figure 2e and f*, *Figure 2—figure supplements 2–6* and *Supplementary file 1*—Tables 8–23) (*Mi et al., 2019*; *Watanabe et al., 2019*). For example, at the gene level, association with *LEPR* in these adjusted analysis for HGB elevated to statistical attention (*Figure 2e*). *LEPR* perturbations cause obesity (*Dubern and Clement, 2012*), and LepR$^+$ endothelial niches support HSC survival (*Comazzetto et al., 2021*). Further, adjusted HGB locus-related genes were enriched for some endothelial and mesenchymal development processes, albeit with diminished p-values due to power loss from limited SNP sets (*Figure 2f* and *Supplementary file 1*-Tables 8–17). Adjusted RBC, PLT, and WBC data also demonstrated enrichment of endothelial and cell adhesion pathways (*Figure 2—figure supplement 1* and *Supplementary file 1*-Tables 18–23). These findings highlight the relevance for BMI, WHR, and related biology in regulating multilineage blood traits, including contributions from mesenchyme-derived adipocytes (*Zhong et al., 2020*) and stromal endothelial cells in bone marrow (*Comazzetto et al., 2021*).

## Discussion

The obesity epidemic has increased the importance of understanding associated systemic comorbidities (*Koenen et al., 2021*), including complex physiology linking cardiometabolic and blood traits. While some clinical epidemiological studies have proposed iron deficiency and chronic inflammation to explain anemia observed in obese populations (*Aigner et al., 2014*; *Benova and Tencerova, 2020*; *Koenen et al., 2021*), confounders inherent to observational studies may limit interpretation. Consistent with most clinical observations (*Aigner et al., 2014*), genetically determined BMI is indeed causally associated with lower HGB and HCT levels.

We identified divergent genome-wide effects of BMI and WHR on erythroid traits. Whereas increased BMI or WHR are typically thought to concordantly raise cardiovascular risk, our results agree with recent findings showing that adipose distribution can influence obesity-related comorbidities (*Huang et al., 2021*). We were somewhat surprised to identify impacts for BMI and WHR on multilineage blood traits that extended beyond clinically reported erythroid effects, suggesting BMI and WHR may act through different mechanisms than previously proposed (*Aigner et al., 2014*; *Koenen et al., 2021*). While the absolute effect sizes are unlikely to impact patient management (e.g., a 1 SD unit increase in BMI [~4.8 kg/m$^2$] decreases HGB by ~0.06 g/dL [*Beutler and Waalen, 2006*; *Gharahkhani et al., 2019*]), genetic mechanisms linking BMI, WHR, and blood traits may help elucidate how

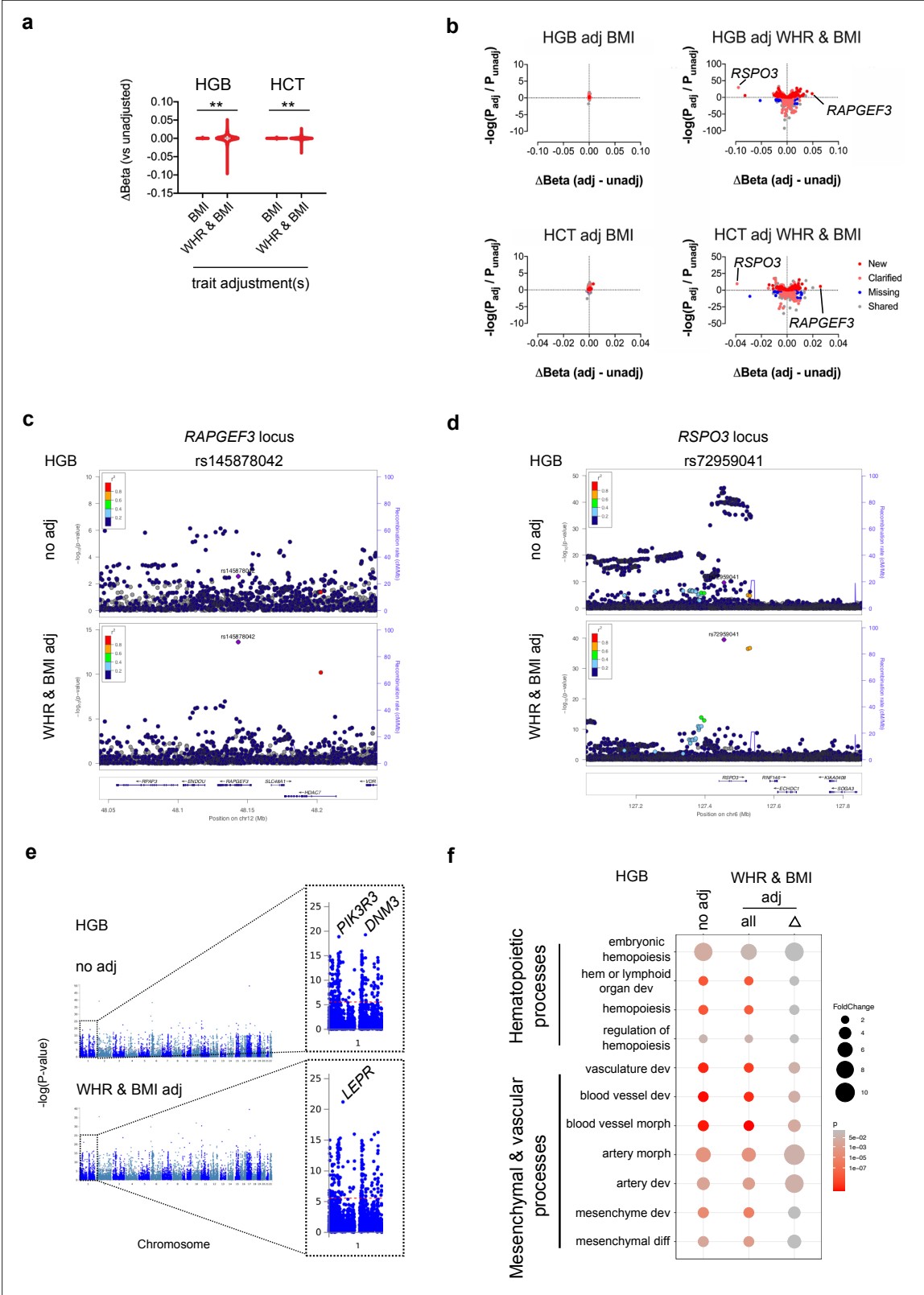

**Figure 2.** Conditional blood trait analysis based on body mass index (BMI) and/or waist-to-hip ratio (WHR) modifies interpretation of genomic loci that impact blood trait variation. (**a**) Violin plots showing the dispersion in effect size at genome-wide significant loci after adjusting erythroid traits (hemoglobin [HGB] or hematocrit [HCT]) for BMI, or WHR and BMI. **p < 0.0001 by F-test to compare variances. (**b**) Scatterplots depicting changes in effect sizes and p-values for all genome-wide significant sentinel loci before or after adjustment. Novel loci (red) had p < 5 × 10⁻⁸ only after adjustment

*Figure 2 continued on next page*

*Figure 2 continued*

and represent new loci (not in LD with genome-wide significant single nucleotide polymorphisms [SNPs] before adjustment). Clarified loci (pink) are sentinel SNPs with $p < 5 \times 10^{-8}$ after adjustment and are in linkage disequilibrium with significant pre-adjustment SNPs. Missing loci (blue) are those with adjusted $p > 5 \times 10^{-8}$, which were significant pre-adjustment. Shared SNPs (gray) are sentinel SNPs before and after adjustment for the indicated factors. (**c**) After adjustment for WHR and BMI, the common coding SNP (rs145878042) in *RAPGEF3* significantly impacts HGB level. (**d**) Adjustment for WHR and BMI alters interpretation of SNP effects at the *RSPO3* locus, including more significant effects for new sentinel variant rs72959041 (unadjusted $p = 2.1 \times 10^{-10}$, adjusted $p = 3.4 \times 10^{-40}$). (**e**) Gene-based Manhattan plots for HGB, before or after BMI/WHR adjustment. (**f**) Gene ontology analyses for hematopoietic, mesenchymal, and vascular biological processes for HGB loci before and after mtCOJO adjustment for BMI and WHR. Significance reflects Fisher's exact test after multiple testing.

The online version of this article includes the following figure supplement(s) for figure 2:

**Figure supplement 1.** Conditional blood trait analysis based on body mass index (BMI) and/or waist-to-hip ratio (WHR) modifies interpretation of genomic loci that impact variation in red blood cell (RBC), platelet (PLT), and white blood cell (WBC) counts.

**Figure supplement 2.** MAGMA tissue enrichment analyses for original and body mass index (BMI)/waist-to-hip ratio (WHR)-adjusted hemoglobin (HGB) data.

**Figure supplement 3.** MAGMA tissue enrichment analyses for original and body mass index (BMI)/waist-to-hip ratio (WHR)-adjusted hematocrit (HCT) data.

**Figure supplement 4.** MAGMA tissue enrichment analyses for original and body mass index (BMI)/waist-to-hip ratio (WHR)-adjusted red blood cell count (RBC) data.

**Figure supplement 5.** MAGMA tissue enrichment analyses for original and body mass index (BMI)/waist-to-hip ratio (WHR)-adjusted platelet count (PLT) data.

**Figure supplement 6.** MAGMA tissue enrichment analyses for original and body mass index (BMI)/waist-to-hip ratio (WHR)-adjusted white blood cell count (WBC) data.

cardiovascular disease (*Heyde et al., 2021*; *Rohde et al., 2021*) and cardiometabolic derangements (*Fuster et al., 2020*; *Jaiswal et al., 2014*) are linked to normal or clonal hematopoiesis.

Directionally consistent effects for BMI and WHR across blood lineages may indicate influences on HSCs in the bone marrow (*Figure 1g*). For example, genetic predisposition to accumulate bone marrow white adipose tissue may underlie age- (*Tuljapurkar et al., 2011*) or obesity-related (*Benova and Tencerova, 2020*) cytopenias by regulating HSC self-renewal or differentiation (*Wang et al., 2018*). Alternatively, genetically determined differences in bone marrow stromal cell types (e.g., mesenchymal stem cell-derived bone marrow adipocytes; *Zhong et al., 2020*) could impact HSC biology. Finally, WHR-related mechanisms impacting blood trait variation may reflect inhibitory paracrine or endocrine effects from gluteal or truncal adipose depots (*Comazzetto et al., 2021*).

Strengths of this study include the use of the largest and most recent GWAS statistics available for all traits, as well as consistent directional effect trends across multiple analyses. However, these analyses were restricted to individuals of European descent, perhaps limiting generalizability. Our study is also subject to limitations of currently available MR methods, including potential MR assumption violations (see Materials and methods), unforeseen pleiotropy, or traits correlating with BMI or WHR that may confound direct causality (*Burgess et al., 2019*).

At minimum, this provides a rationale for concurrent BMI and WHR adjustments when analyzing blood trait GWAS loci to avoid directional bias. These adjustments also provide novel stratification criteria for blood trait GWAS fine mapping studies and candidate blood gene selection. This will be particularly important for studies aiming to explain metabolic or stromal effects on blood cells, which are notably distinct from cholesterol- or lipid-mediated peripheral effects on erythroid cells at the genetic level.

## Materials and methods
### GWAS summary statistics collection

We analyzed publicly available GWAS summary statistics for blood traits (n = 563,085) (*Vuckovic et al., 2020*), BMI (n = 484,680) (*Pulit et al., 2019*), WHR (n = 485,486) (*Pulit et al., 2019*), WHRadjBMI (n = 484,563) (*Pulit et al., 2019*), CAD (n = 547,261) (*van der Harst and Verweij, 2018*), and lipid traits including TC (n = 215,551), TG (n = 211,491), LDL (n = 215,196), and HDL (n = 210,967) (*Klarin et al., 2018*). A glossary of these traits, including unit measurements and descriptions, is

available in *Supplementary file 1*—Table 24. Data were derived from individuals of European ancestry only and were analyzed using genome build hg19/GRCh37.

## Instrumental variable creation

To construct instrumental variables (IVs), we identified all SNPs common to exposure and outcome data sets and clumped genome-wide significant SNPs for the exposure to identify linkage-independent SNPs (EUR $r^2 < 0.01$) in 500 kb regions using TwoSample MR. IV strengths were estimated using F-statistics calculated as described (*Burgess and Thompson, 2011*). IVs used in this study can be found on GitHub (https://github.com/thomchr/ObesityAdiposityBloodMR) or obtained upon request.

## MR and causal effect estimation

Univariable MR analyses (TwoSample MR package v0.5.5; *Hemani et al., 2018*) were conducted using R (v3.6.3). Random variant allele allocation at meiosis enables the MR approach to address confounding and reverse causality that can otherwise preclude causal inference from epidemiologic and cohort studies. Key assumptions must hold in order to make valid conclusions from MR studies. For example, independent genetic instruments (SNPs) must be specifically associated with the exposure trait. Weak instruments, the presence of horizontal pleiotropy, heterogeneity, and error in measured instrument-exposure associations can limit applicability or inferences gleaned from MR studies (*Burgess et al., 2019*).

Presented data show causal estimates from IVW (random effects model), weighted median, and MR Egger regression methods. We assessed pleiotropic bias using MR Egger regression intercepts, which if significantly non-zero can imply directional bias (*Bowden et al., 2015*). MVMR analyses utilized the MVMR package (*Sanderson et al., 2019*) in R. Results shown are IVW method-based causal estimates. Causal direction analyses utilized MR-Steiger and we report values for sensitivity, statistical significance, and inference of the 'correct causal direction' (*Hemani et al., 2017*).

For continuous outcomes (blood traits, lipid traits, BMI), results are presented as beta effect values representing changes in SD units for these traits, per SD unit change in exposure. SD unit estimations were previously calculated for BMI (*Gharahkhani et al., 2019*) and HGB (*Beutler and Waalen, 2006*). For dichotomous outcomes (CAD), causal effect estimates can be converted to odds ratios by exponentiating causal effect estimates ( = exp^[effect]) to calculate a value reflecting the change in outcome per SD unit increase in exposure (*Burgess and Labrecque, 2018*). However, CAD outcome values are presented as SD units to facilitate comparison with blood trait effects.

## Mediation analysis

Mediation analysis estimates were calculated as described (*Burgess et al., 2017*). Total and direct effects are reported for the exposure and mediating trait on each outcome.

## Conditional GWAS analysis

Conditional analyses of filtered SNP sets, containing SNPs found in BMI, WHR, and blood trait summary statistics, were analyzed using mtCOJO with a limit of $r^2 < 0.01$ (*Yang et al., 2011*). Results were clumped using plink (v1.90 beta) (*Purcell et al., 2007*) to identify linkage-independent sentinel SNPs with $r^2 < 0.01$ in 500 kb genomic regions (flagged parameters were `--clump-p1` 5E-8 `--clump-p2` 1 `--clump-r2` 0.01 `--clump-kb` 500). Separate experiments were performed on the same filtered SNP sets to adjust for BMI, or both BMI and WHR. To compare uncorrected with BMI- or BMI-and-WHR-adjusted results, we aggregated sentinel SNPs and clumped based on original GWAS p-values (`--clump-p1` 1 `--clump-p2` 1 `--clump-r2` 0.01 `--clump-kb` 500) to retrieve a complete set of linkage-independent loci. This second clump output allowed us to calculate how many regions were shared, nullified, or novel in the adjusted vs. unadjusted data sets. The gene nearest to each sentinel locus was identified using bedtools (*Quinlan and Hall, 2010*). Locus zoom plots were created through the online instrument (http://locuszoom.org, *Pruim et al., 2010*).

## Gene-level analyses

We identified gene and tissue associations for blood trait summary statistics before and after adjustment using FUMA (*Watanabe et al., 2019*), which uses MAGMA for gene identification (*de Leeuw et al., 2015*).

## Gene ontology

Gene lists were analyzed for significantly over- or under-enriched gene ontology biological processes. Statistical significance was assigned based on Fisher's exact test p < 0.05 after Bonferroni correction for multiple testing (http://geneontology.org, *Mi et al., 2019*).

## Statistical analyses and data presentation

Estimated effects from exposure(s) on outcome are presented from IVW, weighted median, and MR Egger regression measures. Because Cochran's Q test (included in the TwoSample MR package; *Hemani et al., 2018* found heterogeneity in some IVs, we utilized the random effect model when performing inverse variance weighted MR). Thus, we performed and report MR results using IVs that had not undergone pruning. Statistical significance was defined as p < 0.05 for all experiments. For experiments that analyzed 16 blood traits, we also report those that met a more stringent threshold of p < 0.003 (~0.05/16).

Statistics were calculated with GraphPad Prism 8. Figures were prepared using GraphPad Prism 8 and Inkscape (v1.1). Schematic cartoons were created using BioRender.

## Coding scripts and data sets

All relevant coding scripts and data sets can be found on GitHub (https://github.com/thomchr/ObesityAdiposityBloodMR; copy archived at swh:1:rev:4f8e3ae9898f2dcff-2378d02a0977146dc4e0545; *Thom, 2022*). All data and coding scripts are also available upon request.

# Acknowledgements

CST and BFV designed the study. CST, MW, STC, and BFV conducted, analyzed, and/or interpreted experimental data. CST and BFV wrote the paper. All authors approved the final version of the manuscript. This work was supported by National Institutes of Health (T32 HD043021, K99 HL156052, and U24 HL134763 to CST; U01 HL124696 to STC; R56 DK101478 and UM1 DK126194 to BFV), a Linda Pechenik Montague Investigator Award (BFV), and a Children's Hospital of Philadelphia K-readiness award (CST).

# Additional information

### Funding

| Funder | Grant reference number | Author |
|---|---|---|
| Eunice Kennedy Shriver National Institute of Child Health and Human Development | T32HD043021 | Christopher S Thom |
| National Heart, Lung, and Blood Institute | K99HL156052 | Christopher S Thom |
| National Heart, Lung, and Blood Institute | U01HL124696 | Stella T Chou |
| National Institute of Diabetes and Digestive and Kidney Diseases | R56DK101478 | Benjamin F Voight |
| National Institute of Diabetes and Digestive and Kidney Diseases | UM1DK126194 | Benjamin F Voight |
| Linda Pechenik Montague | Investigator Award | Benjamin F Voight |
| Children's Hospital of Philadelphia | K-readiness award | Christopher S Thom |

| Funder | Grant reference number | Author |
|---|---|---|
| National Heart, Lung, and Blood Institute | U24HL134763 | Christopher S Thom |

The funders had no role in study design, data collection and interpretation, or the decision to submit the work for publication.

### Author contributions
Christopher S Thom, Conceptualization, Data curation, Formal analysis, Funding acquisition, Investigation, Methodology, Writing – original draft, Writing – review and editing; Madison B Wilken, Formal analysis, Writing – review and editing; Stella T Chou, Investigation, Writing – review and editing; Benjamin F Voight, Conceptualization, Data curation, Funding acquisition, Investigation, Supervision, Writing – original draft, Writing – review and editing

### Author ORCIDs
Christopher S Thom  http://orcid.org/0000-0003-1830-9922
Benjamin F Voight  http://orcid.org/0000-0002-6205-9994

### Decision letter and Author response
Decision letter https://doi.org/10.7554/eLife.75317.sa1
Author response https://doi.org/10.7554/eLife.75317.sa2

## Additional files

### Supplementary files
- Supplementary file 1. Supplementary tables 1-24.
- Transparent reporting form

### Data availability
The current manuscript is a computational study based on publicly available data sets, so no primary data were generated for this manuscript. All relevant coding scripts and data sets can be found on GitHub (https://github.com/thomchr/ObesityAdiposityBloodMR) or by request.

The following previously published datasets were used:

| Author(s) | Year | Dataset title | Dataset URL | Database and Identifier |
|---|---|---|---|---|
| Vuckovic et al | 2020 | Blood Trait Summary Statistics (UKBB) | ftp://ftp.sanger.ac.uk/pub/project/humgen/summary_statistics/UKBB_blood_cell_traits/ | GWAS Catalog, GCST90002379-GCST90002407 |
| Vuckovic et al | 2020 | Blood Trait Summary Statistics (Meta-analysis) | http://www.mhi-humangenetics.org/en/resources | humangenetics, Meta-analysis |
| Pulit et al | 2019 | Obesity and Adiposity Summary Statistics | https://doi.org/10.5281/zenodo.1251813 | Zenodo, 10.5281/zenodo.1251813 |
| van der Harst and Verweij | 2018 | CAD Summary Statistics | https://data.mendeley.com/datasets/2zdd47c94h/1 | Mendeley Data, 10.17632/gbbsrpx6bs.1 |
| Klarin et al | 2018 | Lipid Trait Summary Statistics | https://www.ncbi.nlm.nih.gov/projects/gap/cgi-bin/study.cgi?study_id=phs001672.v6.p1 | NCBI BioProject, phs001672.v6.p1 |

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
