## [Editor Report]

The study shows that genetically determined adiposity plays a previously underappreciated role in determining blood cell formation and function. The authors have clearly spelled out their hypothesis and performed all the relevant and available analyses in the "Mendelian Randomization toolbox". The study will help understand the pathogenesis for clonal hematopoiesis.

---

## [Decision Letter]

**Decision letter after peer review:**

Thank you for submitting your article "Obesity and adiposity have opposing genetic impacts on human blood traits" for consideration by *eLife*. Your article has been reviewed by 3 peer reviewers, including Sara Hägg as Reviewing Editor and Reviewer #1, and the evaluation has been overseen by David James as the Senior Editor.

In your revision, please address in particular the following issues, as well as the points mentioned in the individual reviewers' reports below.

1) Please explain in more detail why the study was done, what the main hypothesis tested were, and how are all the extra analyses fit into this. There are many different exposures and outcomes used, which makes it difficult to follow the story. For example, how are all the blood traits connected, what were the expected associations and how does the result confirm or contradict those expectations?

2) Both adiposity (WHR) and obesity (BMI) are introduced without further explanations. Why was BMI defined as obesity and WHR as adiposity? In many studies, BMI is referred to as adiposity as well. Especially when a continuous scale of SD increase is used, as in this study. If obesity is what is intended, this is most often defined as a BMI > 30, hence a dichotomized value.

3) Please provide a better definition of some key terms such as genetically determined body mass index.

4) More details are needed around units presented.

5) Is there additional clinical evidence available that supports the findings – for instance are obese people with low hematocrit more or less likely to develop other metabolic diseases?

6) In biological contexts such as here, sex should be used to refer to men versus women.

7) Please discuss the generalizability of your findings as well as strengths and weaknesses of your study.

*Reviewer #1 (Recommendations for the authors):*

Overall, why was the study done? What is the main hypothesis that was tested, and how are all the extra analyses fitting into this? There are many different exposures and outcomes used, and it makes it difficult to follow. In what way are all the blood traits connected? What were the expected associations and how does the result confirm or contradict those expectations?

Related to that, in the introduction both adiposity (WHR) and obesity (BMI) are introduced without further explanations. Why was BMI defined as obesity and WHR as adiposity? No references listed. In many studies, BMI is referred to as adiposity as well. Especially when a continuous scale of SD increase is used, as in this study. If obesity is what is intended, this is most often defined as a BMI > 30, hence a dichotomized value.

More details are needed around units presented. What does an SD increase in BMI causing a 0.057 SD decrease in HGB mean? If this study is indeed done to facilitate interpretation and clinical management as claimed, what is the clinical interpretation of this result? Unit change in clinical meaningful HGB levels, and put in relation to e.g. a treatment effect?

Gender is a socio-cultural concept. Sex is used in the biological context.

No discussion around generalizability or strengths and weaknesses with the current study. MR assumptions or shortcomings with the method not described.

*Reviewer #2 (Recommendations for the authors):*

The study is well performed, presented, written, and novel. The authors utilize summary statistics from publicly available resources.

The methodological section is adequate and clearly describes resources, design, and statistics. The authors performed all the relevant and available MR analyses in the "toolbox".

Figures are good.

Figure 2f: spell error in "process" (spelled "rocess")

Figure supplement 2 a-e: these figures are almost impossible to read. Please make one figure per page.

*Reviewer #3 (Recommendations for the authors):*

The manuscript would benefit from a better definition of some key terms such as genetically determined body mass index, the authors here probably intend to describe the genetic component of BMI as described in M and M. Along the same lines, the authors use truncal adiposity and adipose distribution, rather than adipose tissue or fat.

Lines 136-137, the authors should put the 36 folds changes into appropriate context (e.g. from x% SD to y% SD), otherwise it would be hard for the inexperienced reader to appropriately judge these changes.

Some parts of the discussion (lines 197-199) are hard to follow and perhaps they should be explained with a few more sentences and/or in more layman terms.

In line 204 the authors claim that there is the first example of divergent genome wide effects between BMI and truncal adiposity, however similar claims have been made also in one of the articles from which they took part of the data for their analyses.

---

## [Author Response]

1) Please explain in more detail why the study was done, what the main hypothesis tested were, and how are all the extra analyses fit into this. There are many different exposures and outcomes used, which makes it difficult to follow the story. For example, how are all the blood traits connected, what were the expected associations and how does the result confirm or contradict those expectations?

Thank you for this important recommendation. We have clarified our hypotheses, study structure, and intent in the Introduction (pg 4 lines 72-86). In addition, we added clarifications and subheadings throughout the Results section to help make the story easier to follow. Finally, we have specified in the Discussion which findings confirmed or contradicted our hypotheses (pg 9 lines 212-218).

2) Both adiposity (WHR) and obesity (BMI) are introduced without further explanations. Why was BMI defined as obesity and WHR as adiposity? In many studies, BMI is referred to as adiposity as well. Especially when a continuous scale of SD increase is used, as in this study. If obesity is what is intended, this is most often defined as a BMI > 30, hence a dichotomized value.

Thank you for pointing out this important but confusing terminology. Our intent was to investigate effects of genetically determined BMI or genetically determined WHR as continuous quantitative variables. These key traits are now better defined and explained in the Introduction (pg 3 lines 63-64 and 60-70). We also changed ‘obesity’ to ‘BMI’ and adiposity to ‘WHR’ throughout the text to clarify this issue. We now define clinical obesity in the Introduction (pg 3 line 65), and only reference this in the context of a dichotomized clinical observation to try to avoid/minimize confusion.

3) Please provide a better definition of some key terms such as genetically determined body mass index.

This key term is now better explained in the Introduction. We have also included definitions and units for important terms and traits in Supplementary file 1-Table 24.

4) More details are needed around units presented.

Units and definitions for key terms and traits are now included in Supplementary Tables-Table 24. We have also included some context for the magnitude of effects identified in our experiments (pg 9 lines 218-223).

5) Is there additional clinical evidence available that supports the findings – for instance are obese people with low hematocrit more or less likely to develop other metabolic diseases?

This is an interesting point without exact precedent in the literature. This might be due to clinical confounders that may occur in such individuals. However, recent findings have shown that cardiovascular disease can impact bone marrow hematopoiesis (Heyde *et al., Cell* 2021 and Rhode *et al., Nat Cardiovasc Res* 2021), and cardiometabolic derangements can occur in the context of clonal hematopoiesis (e.g., Jaiswal *et al., NEJM* 2014 and Fuster *et al., Cell Rep* 2020). Our findings may help clarify genetic mechanisms linking these important disease processes (pg 9 lines 220-223).

6) In biological contexts such as here, sex should be used to refer to men versus women.

We have changed ‘gender’ to ‘sex’ throughout the manuscript.

7) Please discuss the generalizability of your findings as well as strengths and weaknesses of your study.

These important points are now included the Discussion (pg 10, lines 235-240). We have also defined assumptions and limitations associated with Mendelian randomization studies in the Methods section (pg 11-12 lines 270-276).

Reviewer #1 (Recommendations for the authors):Overall, why was the study done? What is the main hypothesis that was tested, and how are all the extra analyses fitting into this? There are many different exposures and outcomes used, and it makes it difficult to follow. In what way are all the blood traits connected? What were the expected associations and how does the result confirm or contradict those expectations?

Thank you for this important recommendation. We have clarified our hypotheses, study structure, and intent in the Introduction (pg 4 lines 72-86). In addition, we added clarifications and subheadings throughout the Results section to help make the story easier to follow. Finally, we have specified in the Discussion which findings confirmed or contradicted our hypotheses (pg 9 lines 212-218).

Related to that, in the introduction both adiposity (WHR) and obesity (BMI) are introduced without further explanations. Why was BMI defined as obesity and WHR as adiposity? No references listed. In many studies, BMI is referred to as adiposity as well. Especially when a continuous scale of SD increase is used, as in this study. If obesity is what is intended, this is most often defined as a BMI > 30, hence a dichotomized value.

Thank you for pointing out this important but confusing terminology. Our intent was to investigate effects of genetically determined BMI or genetically determined WHR as continuous quantitative variables. These key traits are now better defined and explained in the Introduction (pg 3 lines 63-64 and 60-70). We also changed ‘obesity’ à ‘BMI’ and adiposity à ‘WHR’ throughout the text to clarify this issue. We now define clinical obesity in the Introduction (pg 3 line 65), and only reference this in the context of a dichotomized clinical observation to try to avoid/minimize confusion.

More details are needed around units presented. What does an SD increase in BMI causing a 0.057 SD decrease in HGB mean? If this study is indeed done to facilitate interpretation and clinical management as claimed, what is the clinical interpretation of this result? Unit change in clinical meaningful HGB levels, and put in relation to e.g. a treatment effect?

Thanks for this recommendation. We have included additional details around units presented (Supplementary Tables-Table 24) and clarified the discussion of clinical relevance. Our findings are most relevant for the discovery of genes and related mechanisms (pg 9 lines 218-223).

Gender is a socio-cultural concept. Sex is used in the biological context.

Thank you for this recommendation. We changed ‘gender’ to ‘sex’ throughout the paper.

No discussion around generalizability or strengths and weaknesses with the current study. MR assumptions or shortcomings with the method not described.

These important points are now included the Discussion (pg 10, lines 235-240). We have also defined assumptions and limitations associated with Mendelian randomization studies in the Methods section (pg 11-12 lines 270-276).

Reviewer #2 (Recommendations for the authors):The study is well performed, presented, written, and novel. The authors utilize summary statistics from publicly available resources.The methodological section is adequate and clearly describes resources, design, and statistics. The authors performed all the relevant and available MR analyses in the "toolbox".Figures are good.Figure 2f: spell error in "process" (spelled "rocess")

Thank you for identifying this error, which is corrected in our revised manuscript.

Figure supplement 2 a-e: these figures are almost impossible to read. Please make one figure per page.

Thanks for this suggestion, each panel is now a separate figure supplement.

Reviewer #3 (Recommendations for the authors):The manuscript would benefit from a better definition of some key terms such as genetically determined body mass index, the authors here probably intend to describe the genetic component of BMI as described in M and M. Along the same lines, the authors use truncal adiposity and adipose distribution, rather than adipose tissue or fat.

Throughout the revised manuscript, we better define these key terms and provide more specific reference to measured traits. As you correctly suggest, the relevant traits include the genetic component of BMI and truncal adipose distribution, as measured by WHR.

Lines 136-137, the authors should put the 36 folds changes into appropriate context (e.g. from x% SD to y% SD), otherwise it would be hard for the inexperienced reader to appropriately judge these changes.

Thanks for this suggestion, we have now included these values (pg 7 lines 162-163).

Some parts of the discussion (lines 197-199) are hard to follow and perhaps they should be explained with a few more sentences and/or in more layman terms.

Thanks for this recommendation, we have reorganized and better explained this part of the Discussion (pg 9-10 lines 225-233). In the interest of clarity, we removed the speculative statement you mention. This was about how decreased hip circumference could impact blood count variation. Future work will test this hypothesis.

In line 204 the authors claim that there is the first example of divergent genome wide effects between BMI and truncal adiposity, however similar claims have been made also in one of the articles from which they took part of the data for their analyses.

Thank you for pointing this out. We have clarified this statement in the revised manuscript (now pg 9 lines 213-215).